# The Dynamics of the Neuropeptide Y Receptor Type 1 Investigated by Solid-State NMR and Molecular Dynamics Simulation

**DOI:** 10.3390/molecules25235489

**Published:** 2020-11-24

**Authors:** Alexander Vogel, Mathias Bosse, Marcel Gauglitz, Sarah Wistuba, Peter Schmidt, Anette Kaiser, Vsevolod V. Gurevich, Annette G. Beck-Sickinger, Peter W. Hildebrand, Daniel Huster

**Affiliations:** 1Institute for Medical Physics and Biophysics, University of Leipzig, Härtelstr. 16-18, D-04107 Leipzig, Germany; alexander.vogel@medizin.uni-leipzig.de (A.V.); mathiasbosse.mb@gmail.com (M.B.); marcel.gauglitz@fu-berlin.de (M.G.); S-Wistuba@t-online.de (S.W.); peter.schmidt@medizin.uni-leipzig.de (P.S.); 2Faculty of Life Sciences, Institute of Biochemistry, University of Leipzig, Brüderstr. 34, D-04103 Leipzig, Germany; anette.kaiser@uni-leipzig.de (A.K.); abeck-sickinger@uni-leipzig.de (A.G.B.-S.); 3Vanderbilt University Medical Center, 2200 Pierce Avenue, Nashville, TN 37232, USA; vsevolod.gurevich@vanderbilt.edu

**Keywords:** GPCR, arrestin, molecular switch, NMR spectroscopy, structural dynamics, MD simulation

## Abstract

We report data on the structural dynamics of the neuropeptide Y (NPY) G-protein-coupled receptor (GPCR) type 1 (Y1R), a typical representative of class A peptide ligand GPCRs, using a combination of solid-state NMR and molecular dynamics (MD) simulation. First, the equilibrium dynamics of Y1R were studied using ^15^N-NMR and quantitative determination of ^1^H-^13^C order parameters through the measurement of dipolar couplings in separated-local-field NMR experiments. Order parameters reporting the amplitudes of the molecular motions of the C-H bond vectors of Y1R in DMPC membranes are 0.57 for the Cα sites and lower in the side chains (0.37 for the CH_2_ and 0.18 for the CH_3_ groups). Different NMR excitation schemes identify relatively rigid and also dynamic segments of the molecule. In monounsaturated membranes composed of longer lipid chains, Y1R is more rigid, attributed to a higher hydrophobic thickness of the lipid membrane. The presence of an antagonist or NPY has little influence on the amplitude of motions, whereas the addition of agonist and arrestin led to a pronounced rigidization. To investigate Y1R dynamics with site resolution, we conducted extensive all-atom MD simulations of the apo and antagonist-bound state. In each state, three replicas with a length of 20 μs (with one exception, where the trajectory length was 10 μs) were conducted. In these simulations, order parameters of each residue were determined and showed high values in the transmembrane helices, whereas the loops and termini exhibit much lower order. The extracellular helix segments undergo larger amplitude motions than their intracellular counterparts, whereas the opposite is observed for the loops, Helix 8, and termini. Only minor differences in order were observed between the apo and antagonist-bound state, whereas the time scale of the motions is shorter for the apo state. Although these relatively fast motions occurring with correlation times of ns up to a few µs have no direct relevance for receptor activation, it is believed that they represent the prerequisite for larger conformational transitions in proteins.

## 1. Introduction

Molecular dynamics is the dominating principle of all living systems. Tissues, cells, membranes, and individual molecules are highly mobile on a broad range of time scales, which provides specific tissue properties [1], allows for cells to adapt to various environments [2], helps maintain membrane elasticity and flexibility [3], is required for proper protein function [4,5], and represents the basis for structural transitions through which proteins convert from the ground state to the activated states [6]. G-protein-coupled receptors (GPCRs) represent a class of particularly mobile molecules. These receptors are localized in the lipid membrane and convert extracellular chemical or physical signals into a series of physical dynamical structural alterations of the molecule to elicit a biological response through intracellular signaling. GPCRs are of fundamental importance in many biological signal transduction cascades and, consequently, highly pharmacologically relevant.

The last two decades have seen tremendous progress in the structure determination of GPCRs [7,8,9]. X-ray and cryo-EM structures have characterized in detail the conformational differences between the apo, agonist-, and antagonist-bound states of various receptors as well as GPCR complexes with G-proteins or arrestins [10,11,12,13,14,15,16,17,18]. Whereas these structures provide a static picture of the different conformations of the receptor in individual states, the dynamics of the structural transitions between such states can be studied in detail by spectroscopic tools [19], in particular by NMR spectroscopy [9]. Using NMR spectroscopy in a solution [20,21,22,23] and in the solid state [24,25], the dynamics of the conformational transitions in GPCRs have been characterized in atomistic detail. Receptor activation is characterized by a seesaw-like swing of transmembrane Helices 6 (TM6) and 7 (TM7), by which the extra- and intracellular ends of the helices are moved in the opposite direction [11]. For the function of GPCRs, a number of activation switches have been identified, which represent well-conserved amino acid residues that change their conformation upon ligand binding, thereby inducing the dynamic reorientation of the TM segments of the molecule [25]. Molecular switches are part of an activation network of interacting residues that undergo restructuring upon activation [26,27]. These conformational transitions occurring on a time scale of milliseconds are observable in NMR spectra by exchange broadening and/or detection of distinct conformations in slow exchange on the NMR time scale [20,21,22,28,29,30].

Whereas each conformational state of a receptor represents a distinct energy well in the complex energy landscape on which GPCRs exist [31], the receptors are also highly dynamic on a fast time scale within the individual energy well of a given state, splitting into conformational substates [32]. Such fast motions occurring with τ < 40 µs are detected as fast fluctuations of bond vectors in the backbone and sidechains, as well as in reorientations of secondary structure elements. Molecular order parameters (*S*) that describe the amplitudes of the motions of a given bond vector (where *S* = 0 means isotropic mobility, whereas *S* = 1 refers to a rigid state) represent a convenient tool to describe these motions in biomolecules [33]. Order parameters are either determined from analysis of spin relaxation rates [34] or by measuring motionally averaged dipolar or quadrupolar couplings [35].

These fast dynamics are also well represented in atomistic molecular dynamics (MD) simulations [36]. While changes between inactive and different active receptor states occur at µs time scales, requiring special-purpose computing systems [37], cloud computing [38], or enhanced sampling methods [39] to explore, short-lived receptor substates can already be sampled at sub-µs time scales. In this way, MD data identify the inherent flexibility of individual segments of GPCRs in agreement with crystallographic B factors [40]. For instance, at equilibrium, the neurotensin receptor shows high root mean square fluctuations for the loop and tail structures and lower values for TM1-7 [40] in agreement with crystallographic B factors. Interestingly, thermostabilization of the neurotensin receptor leads to a significant reduction of these fluctuations [40]. Flexible loops and termini often lack electron density in X-ray structures, suggesting that these segments undergo large amplitude motions or are intrinsically disordered [41,42].

The fast segmental motions of various class A GPCRs reconstituted in lipid membranes have also been probed using solid-state NMR. The most complete data are available for the human neuropeptide Y receptor type 2 (Y2R), which belongs to the neuropeptide Y (NPY) receptor family. NPY receptors play a central role in appetite regulation, anxiety, or maintenance of the circadian rhythm [43]. The crystal structure of the receptor with a small-molecular-weight antagonist has recently been reported [13]. Using uniform ^13^C-labeled Y2R that did not provide site resolution, low average order parameters between 0.55 and 0.67 were determined for the backbone in different liquid crystalline membranes [35,44]. More specifically, site-specific Cα-Hα order parameters for the six Trp residues in Y2R in DMPC membranes prepared by cell-free synthesis ranged from 0.71 to 0.85 in the apo state [24]. The U-^13^C-labeled human growth secretagogue receptor 1a (GHSR) showed similarly high dynamics in membranes, with order parameters between 0.56 and 0.69 [45]. In an effort to increase the site resolution of the NMR studies, the GHSR was prepared by cell-free synthesis with either ^13^C-Met, ^13^C-Arg, or ^13^C-His (representative for the transmembrane domains, the loops and flanking helical regions, or the C-terminus of the receptor) [46]. Although no site resolution was achieved, α-helical residues showed much higher order parameters than the loops.

Here, we report NMR data on the fast dynamics of the neuropeptide Y receptor type 1 (Y1R) in the apo, agonist-, and arrestin-bound states in different lipid membrane environments using solid-state NMR. We complement the experimental work with two sets of long MD simulations of the receptor in the apo and an antagonist-bound state in POPC membranes, providing site-specific information on Y1R dynamics in equilibrium. Our findings refer to a functional role of fast dynamics for ligand binding and downstream signaling in agreement with previous observations.

## 2. Results

### 2.1. Static ^15^N-NMR Spectroscopy on Y1R in Liquid Crystalline Membranes

Static ^15^N-NMR experiments provide an overview of the distribution of rigid and highly mobile segments [44,47]. Molecular motions with correlation times shorter than a few tens of microseconds scale down the ^15^N chemical shift anisotropy, yielding static ^15^N-NMR spectra with a reduced width. First, static ^15^N-MR spectra are sensitive to global motions of the membrane protein in the bilayer. The σ_zz_ element of the ^15^N-CSA tensor is slightly inclined by ~15° with respect to the ^15^N–^1^H amide bond, yielding static ^15^N-NMR spectra with an anisotropy parameter of η = 0.15. Axially symmetric motions of a membrane protein in the bilayer yield axially symmetric (η = 0) ^15^N-NMR spectra [47]. Second, fast segmental fluctuations lead to further averaging of the ^15^N-CSA tensor and can produce very narrow NMR signals.

Static ^15^N-NMR spectra of Y1R reconstituted into DMPC membranes acquired at different cross-polarization (CP) contact times are shown in Figure 1A–C. The ^15^N-NMR spectra are dominated by powder patterns that can be simulated assuming axially symmetric CSA tensors with a span of Δσ = 145–150 ppm, also observed in other membrane-embedded GPCRs and heptahelical membrane proteins [44,47]. We also observed narrow lines at isotropic NMR frequencies at backbone and sidechain chemical shifts. The intensity of these isotropic ^15^N-NMR lines increases upon an increase of the CP contact time.

Quantification of the area underneath the isotropic and anisotropic lines provides an estimation of the ratio of highly mobile and rigid sites of Y1R. The quantification of such NMR spectra, however, requires great care. CP-based NMR spectra are biased by motions, which has to be considered when interpreting NMR spectra of uniformly labeled molecules that do not show spectral resolution. The efficiency of the polarization transfer from ^1^H to the X nucleus depends on the strength of the dipolar coupling as well as the relaxation time in the rotating frame (*T*_1ρ_). Thus, the rigid sites of a molecule show a rapid buildup of spectral intensity of the X nucleus due to the strong dipolar coupling, whereas mobile sites with averaged (i.e., smaller) dipolar couplings reach maximum intensity at longer CP contact times. *T*_1ρ_ relaxation times may also vary between rigid and mobile sites, introducing further difficulty in interpreting the NMR spectra. Therefore, when mobile and rigid sites of a molecule are not separated spectroscopically, measurements of the dipolar coupling strength should be made at varying CP contact times, as well as with direct polarization of the X nuclei [44].

We deconvoluted the NMR spectral line shapes to separate the isotropic signals from anisotropic signals, which are plotted as a function of CP contact time in Figure 1D. Spectral intensities were fitted to the *I*–*S* model [48], yielding the true intensity ratio of isotropic-to-anisotropic ^15^N sites in the protein backbone. This analysis revealed that 14% of the backbone segments of Y1R in DMPC undergo large amplitude motions responsible for the narrow ^15^N-NMR lines. The isotropic signals reach their maximum intensity in the CP experiment at a longer contact time of 890 µs compared to the anisotropic signals (610 µs). The isotropic sites show lower dipolar couplings of 1/*T*_IS_ = 3.0 kHz, whereas the sites that show anisotropic spectral intensity are much more strongly coupled (1/*T*_IS_ = 5.9 kHz). The *T*_1ρ_ values are more similar for isotropic (4.0 ms) and anisotropic sites (4.8 ms). Our analysis only considers motions with correlation times faster than ~70 µs. It is possible that slower µs time scale motions that interfere with the decoupling or excitation frequencies could lead to signal losses in the ^15^N NMR spectra [49]. Such motions could not be accounted for in our analysis.

### 2.2. ^13^C-NMR Studies of the Molecular Dynamics of Y1R by DipShift Experiments

The ^13^C-NMR spectra of the reconstituted Y1R in DMPC membranes recorded under magic-angle spinning (MAS) conditions display better resolution and signal dispersion than the ^15^N-NMR spectra due to the higher gyromagnetic ratio and the larger chemical shift range of the ^13^C nuclei. Though not reaching site resolution, these ^13^C-NMR spectra enable the differentiation of the signals from the aliphatic Cα, CH_2_, and CH_3_ groups.

We used three different excitation schemes to record ^13^C-NMR spectra of Y1R, cross-polarization, direct excitation, and INEPT NMR spectra. Typical NMR spectra are shown in Figure 2A–C. Pronounced differences in these NMR spectra were found indicative of heterogeneously distributed molecular dynamics of the membrane-embedded molecule. Whereas ^13^C CPMAS NMR spectra (Figure 2A) show relatively broad signals with little site resolution, directly excited ^13^C-NMR spectra (Figure 2B) feature more narrow lines with higher intensity, especially in the side chain and ^13^CO regions attributable to mobile sites. ^13^C INEPT NMR spectra (Figure 2C), which detect only highly mobile sites by *J*-coupled polarization transfer, display numerous receptor signals, especially in the aliphatic side chain region. For comparison, a ^13^C INEPT NMR spectrum of pure DMPC-*d*_54_ membranes is shown in Figure 2D to help identify the lipid signals in the INEPT spectrum of the receptor.

The differences in the NMR spectra clearly indicate that the molecular dynamics of Y1R reconstituted in lipid membranes are heterogeneously distributed over the molecule. To record these differences more quantitatively, we used the separated-local-field experiment DipShift [50] for a quantitative comparison of the amplitudes of motion of the mobile and rigid Y1R segments. First, DipShift experiments were performed for Y1R reconstituted into DMPC membranes. As demonstrated for the ^15^N-NMR spectra, cross-polarization NMR spectra are heavily biased by molecular motions [35]. Therefore, ^13^C DipShift spectra were acquired, using either CP excitation with a contact time of 700 µs or direct excitation.

Figure 3 provides a plot of the molecular order parameters of Y1R in different membranes determined from either CP or directly excited DipShift experiments. Order parameters were calculated as the ratio of the measured motionally averaged CH dipolar coupling divided by the full rigid limit dipolar couplings. Rigid limit values determined from DipShift experiments of crystalline amino acids at low temperatures were taken from the literature [51,52]. Larger order parameters are determined from CP-excited DipShift experiments. Under these conditions, predominantly more rigid sites of Y1R are excited. In contrast, directly excited DipShift experiments report the order parameters of all carbons without dynamic bias. Lower order parameters are measured from directly excited DipShift experiments providing the mean order parameter of the receptor backbone and sidechains. Under these conditions, an order parameter of 0.57 is determined for the protein backbone of Y1R in DMPC. Sidechain order parameters for the methylene and methyl segments are 0.37 and 0.18, respectively, determined at 30 °C (Table 1).

For Y1R reconstituted into more physiological monounsaturated lipid membranes, the order parameters are slightly higher (POPC: *S*_Cα_ = 0.64, *S*_CH2_ = 0.35, and *S*_CH3_ = 0.18; POPC/POPS: *S*_Cα_ = 0.60, *S*_CH2_ = 0.48, and *S*_CH3_ = 0.18, determined from directly excited DipShift spectra), although recorded at a slightly higher physiological temperature of 37 °C. We also reconstituted Y1R into a more complex neuronal lipid mix of POPC/POPE/POPS/cholesterol [53], where slightly higher order parameters were measured (*S*_C__α_ = 0.67, *S*_CH2_ = 0.52, and *S*_CH3_ = 0.21, determined from directly excited DipShift spectra). All order parameter values for Y1R in monounsaturated membranes are given in Table 2.

The first crystal structure of Y1R in the presence of the small antagonist UR-MK299 was recently reported [13]. We measured the NMR order parameters of Y1R in POPC membranes in the presence of this antagonist. These order parameters were similar to those obtained for the apo state of Y1R in POPC membranes (Table 2).

### 2.3. Molecular Dynamics of Y1R in the Presence of the Agonist and Coupled to Arr3-3A

The activation of a GPCR is accompanied by characteristic changes in the energy landscape of these proteins [54], resulting in dynamic alterations. In addition to the characteristic changes observed upon activation and G-protein or arrestin binding [20,21,22,23,24], the equilibrium dynamics of a GPCR is subject to changes [35,44,45]. Here, we probed how the fluctuations of Y1R reconstituted into DMPC membranes would change upon agonist binding and subsequent interaction with arrestin. Order parameters of Y1R reconstituted into DMPC membranes in the absence and presence of NPY and bound to arrestin are shown in Figure 4. ^1^H-^13^C order parameters were measured using excitation by CP (700 µs contact time) as well as by direct polarization. In the presence of NPY, most segments show slightly increased order parameters. In the presence of NPY and Arr3-3A, however, almost all order parameters are higher, suggesting a more constraint equilibrium dynamics of Y1R when bound to arrestin. This trend is particularly clear for the protein backbone.

### 2.4. Molecular Dynamics Simulations of Y1R in the Absence and in the Presence of the Antagonist UR-MK299

To investigate the time-resolved motion of Y1R at atomic resolution, a set of six extensive MD simulations was conducted, starting from the available X-ray structure coordinates with a length of 20 μs each (with the exception of one trajectory that was 10 μs long). Three replicas of MD simulations were started for the antagonist UK-MK299-bound state and apo state of Y1R, respectively. The apo Y1R state was obtained by the removal of UK-MK299, followed by an exceptionally long equilibration of the system of 5.5 μs to allow the receptor to leave the energy minimum of the antagonist-bound state. Each production run was simulated for 20 μs, with the exception of Run 3 of the apo state, which was simulated for 10 μs.

The trajectories were analyzed, and the derived DipShift order parameters were compared to the experimental data applying an established protocol [55]. The backbone C-H order parameters *S*_DipShift_ for each amino acid obtained from MD simulations are only slightly lower than the average order parameter observed in the experiment, presumably reflecting minor differences in system setups (Figure 5A) or insufficient sampling of receptor reorientation as a whole.

The strength of the MD simulations is the site resolution that goes beyond the current experimental data set. From the trajectories obtained, a detailed analysis of the dynamics of Y1R in the absence and presence of UR-MK299 was conducted. To disentangle the nonrelevant overall motions of the receptor from the relevant internal motions, each trajectory frame was aligned to its starting structure to eliminate the dynamics resulting from translational and rotational movements of the protein within the membrane bilayer [56]. The internal order parameters *S*_internal_ were finally calculated as the average value of DipShift order parameters of three runs for each system. The specific dynamics of the respective structural elements identify transmembrane helical segments having higher order than the loops, termini, and Helix 8 (Figure 5B). Mapping the order parameters onto the receptor structure reveals small differences in order between the apo and antagonist-bound states (Figure 5C,D).

For further analysis, the receptor was divided into subsegments to separately quantify the movements of these individual segments. The seven TMs were split in the middle into an extracellular and an intracellular part, exactly where six of the seven TMs feature a kink. The resulting sections are listed in Table 3 and shown in Figure 6 with extracellular helix segments colored in red and intracellular helix segments in blue.

The axis of these helix parts was determined by fitting a vector through the Cα-positions of the helix. Loops, termini, and Helix 8 (due to partial unfolding in some instances) were analyzed in a similar fashion. Since fitting the Cα positions of the whole loop would lead to a vector that mostly points from the end of one helix to the end of the next, each loop was split into at least two parts. This also allows keeping the individual segments similar in length to facilitate comparison (see Table 3).

The vectors obtained from fitting were analyzed by calculation of their order parameters and P_2_ autocorrelation functions (ACF) at a 1 ns time resolution. The order parameters *S*_total_ of the individual segments are shown in Figure 7.

As expected, transmembrane helices have a much higher order than the loops or termini. No clear differences in order between the apo and the antagonist-bound state are found. Somewhat surprising was the observation that all extracellular helix segments have a lower order than their intracellular counterparts, except for Helix 7. The mean order parameter of all extracellular helix segments (UR-MK299-bound state: 0.975, apo state: 0.972) is lower than that of the intracellular helix segments (UR-MK299-bound state: 0.990, apo state: 0.989). This difference in order between the extracellular and intracellular segments is reversed for the loops, termini, and Helix 8, where the mean order parameter of all extracellular segments (UR-MK299-bound state: 0.825, apo state: 0.813) is higher than that of the intracellular segments (UR-MK299-bound state: 0.632, apo state: 0.689). This difference is mostly due to the low order of the ICL3 and the C-terminus.

Using the individual ACFs of the segment vectors, we further split the order parameters into contributions from fast and slow motions. For the sake of this analysis, any motion faster than 1 ns was considered fast. In the ACF, it is represented by a drop from a value of 1 at its start to some lower value at the next data point at a delay of 1 ns. This value of the ACF corresponds to the square of the fast order parameter *S*_fast_. Assuming that the fast motions are independent of the slow motions, the order parameter of the slow motions *S*_slow_ was extracted via [57,58,59]:*S*_total_^2^ = *S*_slow_^2^ × *S*_fast_^2^

The values of the obtained slow and fast order parameters are shown in Figure 7. We observe that the main contributions to reduction in order originate from motion significantly slower than 1 ns. The only exceptions are the extracellular segments of Helices 3 and 4, where the majority of the order reduction is due to contributions from fast motions. These two helix segments are relatively small but do not unfold during the MD simulations.

Furthermore, we used the ACF to determine the correlation times of the motions. For this, we fitted the ACF with a monoexponential decay. The average correlation times for the individual segments are shown in Figure 8.

Here, a clear difference between the simulations of the apo and the antagonist-bound state is observed. There is a clear trend that the apo state shows shorter correlation times in simulations than the antagonist-bound state, by up to a factor of four. The arithmetic mean correlation time of all transmembrane helix segments of the antagonist-bound state (1815 ns) is significantly higher (*p* < 0.01) than that of the apo state (1182 ns). The same trend (*p* < 0.05) is observed for loops, termini, and Helix 8 (UR-MK299-bound state: 1899 ns, apo state: 1507 ns).

Comparing the mean correlation times between the extra- and intracellular segments, no significant differences are observed for the helices (apo state: extracellular: 1195 ns, intracellular: 1169 ns; UR-MK299-bound state: extracellular: 1893 ns, intracellular: 1738 ns). Loops, termini, and Helix 8, however, show some small differences between the extra- and intracellular segments (apo state: extracellular: 1395 ns, intracellular: 1746 ns; UR-MK299-bound state: extracellular: 1822 ns, intracellular: 2023 ns), where the extracellular segments show slightly shorter correlation times than the intracellular segments.

## 3. Discussion

GPCRs are flexible molecules that undergo complex rearrangements in the course of activation [36]. The physical basis for this dynamic is a complex energy landscape on which GPCRs exist with defined energy wells for the ground, intermediate, and activated states separated by defined energy barriers [31,54]. Whereas the individual energy wells represent a distinct state of a receptor, NMR work has shown that the receptors are also subject to relatively large amplitude fluctuations within a specific state, i.e., within a given energy well [35,44,45,46]. Although no site resolution was achieved in these studies, the remarkable conclusion was that GPCRs are subject to more pronounced backbone fluctuations than observed for other membrane proteins of comparable sizes [34,51,60,61,62]. In that regard, Y1R is no exception.

Static ^15^N-NMR spectra revealed that 14% of the backbone residues undergo large amplitude fluctuations, giving rise to very narrow NMR signals. These can be mostly attributed to the long tails of the molecule, which also did not show electron density in the X-ray structure (depending on the antagonist, only Residues 31–339 or 18–337 are resolved) [13]. These tail ends very likely undergo large amplitude motions.

To distinguish between backbone and sidechain order parameters, ^13^C MAS NMR studies using different excitation schemes favoring the detection of rigid or mobile sites, respectively, were employed. We measured the order parameters using either CP excitation or direct excitation to differentiate between the molecular mobility of the more rigid segments of Y1R. ^13^C-NMR spectra provided order parameters characteristic of receptor segments undergoing relatively large amplitude fluctuations on average in different membrane systems. Order parameters determined by CP excitation with a contact time of 700 µs were between 14 and 21% higher than those detected with direct excitation (Table 1 and Table 2). Under the latter conditions, backbone order parameters amounted to surprisingly low values between 0.57 and 0.67 depending on the host membrane, corresponding to remarkable backbone motional amplitudes of the C–H bond vectors of 47° to 40°. In a recent study, the site-specific order parameters of all Trp residues, mostly residing in α-helical secondary structures of the Y2R in DMPC membranes, were measured site-specifically using CP excitation with a 700 µs contact time [24]. This study reported Trp order parameters between 0.71 to 0.85 in the apo state. These values agree well with what we measure for the Y1R using CP excitation, highlighting the transmembrane segments (Table 1). Lower order parameters are measured when NMR spectra were directly excited in agreement with our previous reports [35,44,45,46]. Due to the lack of site resolution in the ^13^C-NMR spectra, a more specific discussion of local differences in backbone fluctuation amplitude is not possible at this stage. Site-specific information, however, is available from the MD simulation (vide infra).

Small alterations in the backbone fluctuations of Y1R are observed in different membrane environments. Generally, membranes composed of longer lipid chains render Y1R more rigid, in accordance with a putative extension of the (more rigid) α-helical secondary structures the receptor may assume when reconstituted into membranes with higher hydrophobic thickness to avoid an energetically unfavorable hydrophobic mismatch as reported for bovine rhodopsin [63]. This effect is stronger in the presence of cholesterol (Figure 3), which condenses lipid chains leading to the increased hydrophobic thickness of the host membrane [64].

Finally, alterations were observed in the overall mobility of Y1R in the presence of a small-molecular-weight antagonist, the agonist, or in complex with Arr3-3A (Figure 4). Here, we used the phosphorylation-independent arrestin variant Arr3-3A [65]. Especially for the protein backbone, a small increase in order was observed upon NPY binding, and a pronounced rigidization was measured upon subsequent Arr3-3A binding. This suggests that the receptor assumes an overall more rigid conformation in complex with arrestin. Our recent study on the Y2R also confirms that the Y2R in complex with Arr3-3A predominantly assumes a single conformation, concluded from the distinct chemical shifts observed for five out of six Trp residues in the molecule [24].

With regard to the correlation times of this receptor dynamics, measurements of motionally averaged dipolar couplings do not provide direct information. All motions with correlation times faster than ~40 µs scale down ^1^H-^13^C dipolar couplings as measured, for instance, in DipShift experiments. Intermediate time scale motions (µs correlation times) are more difficult to detect but give rise to DipShift dipolar dephasing curves that decrease in signal intensity over one rotor period [60,66]. Such dephasing curves were not observed in our DipShift experiments, suggesting that intermediate time scale motions do not contribute significantly to the mobility of Y1R.

Taken together, the experimental part of the study confirms that Y1R is a highly mobile molecule in lipid membranes that can adapt to the specific membrane environment and undergoes a rigidization upon agonist binding and complex formation with Arr3-3A.

To investigate the dynamics of Y1R with full site resolution, we conducted MD simulations in the apo and the antagonist-bound states over more than 100 µs. Such long time scales are necessary, first to be comparable to the NMR time scale, which in the case of the DipShift experiment, has an upper limit of ~40 μs. Second, it was shown that even for motions occurring on the ps to ns time scale, simulation times in the order of several μs are necessary to achieve good agreement between simulation and experiment [67]. The observed order parameters are, on average, somewhat lower than the experimental values but show clear differences between loop and helix segments, as observed experimentally. To further analyze the contributions from helices, loops, and termini, we segmented the receptor into subsegments, with each analyzed independently. Interestingly, extracellular TM helix segments show lower order than their intracellular counterparts. This can also be seen in the profiles of the internal DipShift order parameters (Figure 5B), where within a transmembrane helix, the order parameter slightly drops towards the extracellular side, most significantly for Helices 1 to 5.

The opposite is observed for the loops, where extracellular loops mostly show rather high order, with the exception of Segments 2–4 of the long ELC2 and the first segment of the N-terminus. The intracellular loops show smaller order in general, with ICL3 and the C-terminus having particularly low order. Even the most ordered loop on the intracellular side (ICL1) has lower order than most extracellular segments. Helix 8, which is localized on the intracellular side, has much lower order than any transmembrane helix and lower order than most extracellular segments. This is partly due to the observed tendency to unfold in our simulations. In addition, it is known for other receptors that ICL3 is very flexible [39,68] and Y1R is no exception. In our simulations, the relatively long intracellular ends of Helices 5 and 6 that are connected by ICL3 partly unfold, further increasing its flexibility.

In prototypical receptors, the high flexibility of ICLs was assigned a function for the recognition of intracellular binding partners. In rhodopsin, where signal velocity ensures the role of vision as a central control element in behavior, the flexibility of the ICLs may promote fast signal transfer from rhodopsin to G_t_ through a stepwise and mutual reduction of the conformational space along a common binding funnel. The intrinsically unstructured nature of ICL3 would maximize the capture radius to accelerate the encounter with its binding partner [69]. The high flexibility of the ICL3 interconnecting TM5 and 6, observed in inactive and active receptor states, facilitates binding of the β2-adrenoceptor to G_s_ and G_i_ proteins as the position of TM6 is a major determinant of receptor G-protein coupling specificity [39]. A recent combination of NMR spectroscopy and MD simulations has shown that the unstructured ICL2 of the β2-adrenoceptor only adopts a helical conformation in complex with G_s_ but not with G_i_, underscoring the importance of structural flexibility of intracellular structural elements for specific signaling [70]. In any case, the intrinsic propensity to unfold opens the possibility of rapid dissociation after signal transfer, because, thermodynamically, the formation of secondary structures with minimized degrees of conformational freedom constitutes an entropic cost, which lowers the overall binding affinity.

As determined experimentally, no significant differences in order between the apo and the antagonist-bound states were observed. On the µs time scale, however, the correlation times of the motions are different between the two states. For NMR measurements, this means that investigations on the time scales of motion (e.g., via relaxation measurements) could reveal larger differences between GPCRs bound to different binding partners than investigations on the amplitudes of motions (e.g., via order parameters). For Y1R, the apo state shows considerably shorter correlation times in general. Since both states showed no difference in order, it seems reasonable to assume that both sample a similarly diverse set of structures. The difference in correlation times, however, could mean that the antagonist-bound state undergoes fewer structural transitions than the apo state in accordance with the aforementioned observation that binding events are usually accompanied by an increase in enthalpy and a decrease in entropy. Further research is necessary to investigate this intriguing behavior.

In summary, Y1R is a GPCR that shows comprehensive, fast dynamics with motional amplitudes in the backbone on the order of 40° within each specific state of activation, similarly to other class A peptide-binding GPCRs [35,44,45]. NMR analysis lacking site resolution only allows relatively general conclusions. Further research needs to apply specific labeling, feasible when using cell-free expression [24,46]. Very promising is the combination of NMR and MD simulation to provide a detailed atomistic picture of the (sub-) microsecond dynamics of the molecule. Although the relatively fast motions occurring with correlation times of ns to a few µs have no direct relevance for the dynamic equilibrium of the individual receptor states, it is believed that these fast molecular fluctuations represent the prerequisite for larger conformational transitions in proteins and receptor signaling specificity [33,71].

## 4. Materials and Methods

### 4.1. Materials and NPY Synthesis

All chemicals used for the expression of Y1R were purchased from Sigma-Aldrich (Taufkirchen, Germany) and the lipids from Avanti Polar Lipids, Inc. (Alabaster, AL, USA). The ligand porcine-NPY was obtained by solid-phase peptide synthesis, as previously described [72].

### 4.2. Y1R Expression

The preparation of Y1R followed established procedures on the recombinant expression of GPCRs [73]. To produce ^13^C- and ^15^N-labeled Y1R samples, the 381 amino acid WT receptor with a C-terminal 8× His-Tag was expressed in *E. coli* Rosetta (DE3). The protein was expressed in inclusion bodies by fermentation in a modified M9 minimal medium at 37 °C, as described before [73,74]. The sole nitrogen sources for the production of uniformly ^15^N-labeled Y1R samples were ^15^NH_4_Cl and (^15^NH_4_)_2_SO_4_. Uniformly ^13^C-labeled samples of the receptor were obtained by the addition of ^13^C6-d-glucose to the growth medium approximately 30 min prior to induction. After 4 h of cultivation, cells were harvested. Inclusion bodies were isolated, solubilized, and purified, as described elsewhere [75]. This expression strategy yielded ~15 mg/L of Y1R.

### 4.3. Arr3-3A Expression

For dynamic measurements of Y1R in the presence of arrestin, the phosphorylation-independent variant of *bos taurus* arrestin-3 (Arr3-3A) was added. This variant contained three alanine mutations (Ile397Ala, Val398Ala, Phe399Ala) [65]. This modified arrestin-3 was prepared, as described in [76]. Arr3-3A was expressed in *E. coli* Rosetta(DE3) or *E. coli* NiCo21(DE3) cells in LB medium at 26 °C and 150 rpm. Expression was induced by the addition of IPTG to a final concentration of 35 µM at an OD600 of ~1.0 to 1.5. Multistep cell lysis included the addition of lysozyme (Roth, Karlsruhe, Germany), freezing at −80 °C, sonication, incubation with 8 mM MgCl_2_ plus DNase (Sigma-Aldrich, Taufkirchen, Germany), and several centrifugation steps. The protein was precipitated by the addition of ammonium sulfate to a final concentration of 2.4 M, pelleted, and dissolved in column buffer. The following chromatography steps included purification on a heparin-Sepharose column, Q- and SP-Sepharose columns (GE Healthcare). The purification steps were validated by SDS-PAGE and Western blot.

### 4.4. Y1R Sample Preparation

Then Y1R was solubilized in 50 mM sodium phosphate buffer (15 mM SDS at pH 8.0) at a concentration of 0.5 mg/mL and dialyzed against 50 mM sodium phosphate buffer (2 mM SDS, 2 mM reduced glutathione (GSH), and 1 mM oxidized glutathione (GSSG) at pH 8.5) for the formation of the disulfide bridge [77]. Subsequently, the Y1R was transferred into 50 mM sodium phosphate buffer (1 mM EDTA at pH 8.0) containing the respective phospholipid and DHPC-c7 at a molar ratio of 200:1200:1 (phospholipid/DHPC/Y1R). Bicelle formation was achieved by three freeze–thaw cycles at 0 °C and 42 °C, respectively. Afterwards, the receptor solution was added to the bicelle mixture, followed by three additional cycles from 42 to 0 °C [78]. Reduction of the DHPC concentration resulting in the formation of larger bicelles was obtained by adding 75 mg/mL BioBeads (Bio-Rad, Feldkirchen, Germany) to the protein–lipid mixture twice. Biobeads were removed, and the sample was pelleted by ultracentrifugation at 86,000× *g* and filled into MAS NMR rotors for NMR measurements. For NPY and NPY/Arr3-3A-containing samples, NPY was added in fourfold and Arr3-3A in twofold excess prior to the final centrifugation step. For samples prepared in the presence of Arr3-3A, slightly modified buffer conditions were applied (50 mM sodium phosphate, 1 mM EDTA, 200 mM NaCl, pH 8).

### 4.5. NMR Experiments

Static ^15^N CP NMR spectra were acquired on a Bruker Avance I 750 MHz NMR spectrometer using a double-channel probe with a 5 mm solenoid coil. After cross-polarization of the ^15^N-nuclei with varying CP contact times, the NMR signal was acquired by Hahn echo detection under TPPM decoupling with an RF field strength of 62.5 kHz [79]. The ^15^N-NMR spectra were simulated numerically for deconvolution of the axially symmetric powder pattern and the narrow peaks resulting from rigid and flexible ^15^N nuclei, respectively [44]. These points were fitted to a CP-build-up curve [48].

The ^13^C MAS NMR experiments were performed on Bruker Avance III 600 and Avance Neo 700 NMR spectrometers using a double resonance magic-angle spinning (MAS) probe equipped with 3.2 mm or 4 mm spinning modules. The pulse lengths for 90° pulses for ^1^H and ^13^C were 4 and 5 µs, respectively. Standard CPMAS- and INEPT NMR experiments were acquired at a MAS frequency of 7 kHz using Spinal decoupling at RF fields of 50 and 21 kHz, respectively.

Constant time ^1^H-^13^C DipShift experiments [80] were recorded by detecting the time evolution of the ^1^H-^13^C dipolar coupling over one rotor period at a MAS frequency of 5 kHz. The excitation for the ^13^C nuclei was achieved either by direct excitation or by cross-polarization with contact times of 700 and 2000 µs. During *t*_1_ evolution, homonuclear decoupling was applied by the FSLG-sequence with an effective field strength of 80 kHz [81]. The dipolar dephasing curves were simulated as described before [44], and the obtained dipolar couplings were divided by the known rigid limits (determined from experiments on crystalline amino acids at low temperature) to obtain order parameters [51,52].

### 4.6. MD Simulations

Two different systems were investigated with MD simulations: the apo state and an antagonist-bound (UR-MK299) state of Y1R. For the structure of Y1R, the published crystal structure (PDB ID: 5ZBQ) was used in both systems [13]. In the crystal structure, Phe129 was mutated to Trp, which was reverted to in the MD simulations. In addition, ICL3 was missing from the crystal structure and built using SuperLooper2 [82]. Two amino acids at the C-terminus were added using the PyMOL molecular graphics system, version 2.3.2 Schrödinger, LLC to include the palmitoylation at Cys338. Hydrogen atoms were added to the protein structure, and the N- and C-termini were capped with the patches ACE and CT1 from the CHARMM force field, respectively [83]. Water molecules from the crystal structure were retained, and any remaining receptor cavities were filled with additional water using dowser [84]. All residues were kept in the standard protonation states of the CHARMM36 force field, with the exception of the highly conserved Asp86, which was protonated in the presence of UR-MK299. For the apo simulations, the antagonist UR-MK299 was removed, Asp86 remained deprotonated, and a sodium ion was placed next to Asp86, as this is known to be present in many inactive structures [85] and Y1R in particular shows an attenuation of agonist binding in the presence of Na^+^ [86,87]. These ions stayed in this position for several microseconds of simulation time but eventually left the receptor interior in all three apo simulations and frequently returned to the receptor interior and, in one case, got stably attached to Asp86 again. For setup of the environment 200 POPC molecules, ~21,000 TIP3 water [88] and 5 (apo) or 6 (UR-MK299) chloride ions (to neutralize the system) were added in a rectangular box of ~87 Å side length (x and y) and ~125 Å height (z) using published procedures [89,90,91,92,93,94,95].

The simulations were run in the NPT ensemble at a temperature of 310.15 K and a pressure of 1.013 bar using GROMACS 2019.4 and newer. The CHARMM36 force field [83] was employed for lipids and proteins. The CgenFF [96] generalized force field was used to describe the antagonist UR-MK299. Particle-mesh Ewald was used to treat electrostatic interactions, using a cut-off distance of 10 Å. Bonds involving hydrogen were constraint with LINCS [97] to allow a time step of 2 fs. Each system containing about 95,000 atoms was energy minimized with the steepest descents algorithm and 1000 kJ mol^−1^ nm^−1^ as the threshold. All systems were equilibrated with harmonic positional restraints applied to lipids and Cα atoms of the protein that were sequentially released in a series of equilibration steps. For each system (apo and UR-MK299), a total of three MD simulations was run. For the apo system, additional very long unbiased equilibration times were used (Run 1: 6.36 μs, Run 2: 6.31 μs, Run 3: 5.49 μs) to allow adopting the structure to the absence of the antagonist. For the UR-MK299 system, considerable time was spent on unbiased equilibration (Run 1: 2.05 μs, Run 2: 1.88 μs, Run 3: 1.86 μs) to allow the relaxation of crystal contacts. All production runs were simulated for 20 μs, with the exception of apo Run 3, which was simulated for 10 μs. MDsrv sessions [98] of Run 1 of both systems are available under http://proteinformatics.org/mdsrv.html?load=file://public/papers/y1_dynamics/apo.ngl and http://proteinformatics.org/mdsrv.html?load=file://public/papers/y1_dynamics/ur-mk299.ngl.

For the analysis of the MD simulations, DipShift order parameters were calculated following a published procedure [55]. Further, different segments were defined, and their orientations at each time point were determined by fitting the Cα positions. For loops, termini, and Helix 8, a regular fit was conducted, where the sum of the squared distances *d_i_* of the Cα positions from the orientation vector was minimized. In the case of TM helices, a slightly adjusted approach was used, where the mean distance $\bar{d}$ of all Cα positions from the orientation vector was determined and the sum of the squared deviations of the individual distances from this mean $\underset{i}{\sum }{\left(d_{i}-\bar{d}\right)}^{2}$ was minimized. This way, the typical tilt of the orientation axis due to the residues at the ends of the helix was avoided. The order parameter of each segment was calculated from its orientation axis. First, the mean orientation vector $\bar{v}$ was calculated, and then, the order parameter *S*_total_ was calculated from the orientations *v_t_* at each time step *t* via $S_{\mathrm{total}}=\langle 3{\left(v_{t}/\left|v_{t}\right|\cdot \bar{v}/\left|\bar{v}\right|\right)}^{2}-1\rangle_{t}/2$, where the angle brackets denote the average over all *t*. The ACFs of the orientation vectors were calculated in similar fashion, where for each time delay *dt*, the value of the ACF at this delay was calculated as $\mathrm{ACF}\left(dt\right)=\langle 3{\left(v_{t}/\left|v_{t}\right|\cdot v_{t+dt}/\left|v_{t+dt}\right|\right)}^{2}-1\rangle_{t}/2$. In the analysis, the ACFs were fitted by a monoexponential decay $\mathrm{ACF}\left(dt\right)=A\cdot e^{-dt/\tau }+S_{\mathrm{total}}^{2}$, where the known order parameter *S*_total_ was used to reduce the number of fitting parameters to two (*A*: amplitude of the function, τ: correlation time). For these fits, only the first third of the ACFs was used, as they tend to become very noisy at long time delays *dt*. For statistical analysis, we used the two-sample *t*-test using summarized data, with the Welch correction applied (using OriginPro 2017), for comparison of the correlation times of the segments in two different GPCR states in Figure 8. For analyzing the statistical significance between a number of segments, we used the pair-sample *t*-test using raw data (using OriginPro 2017), where the individual runs were paired with each other (e.g., Run 1 of a segment in the apo state vs. Run 1 of the same segment in the antagonist-bound state).

## Figures and Tables

**Figure 1 molecules-25-05489-f001:**
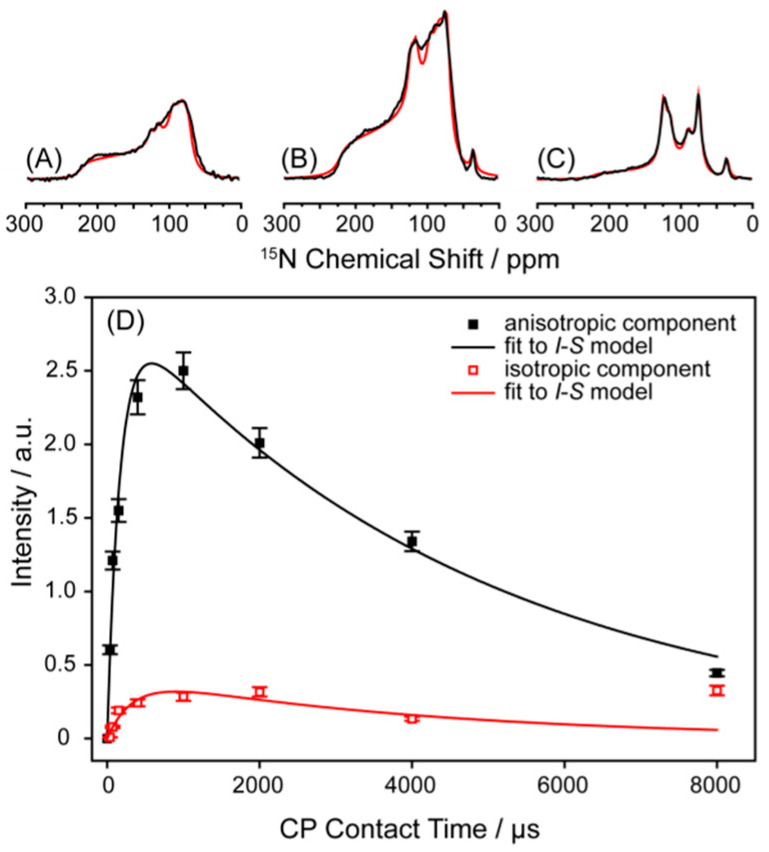
Proton-decoupled static ^15^N-NMR spectra of uniformly ^15^N-labeled neuropeptide Y receptor type 1 (Y1R) in DMPC membranes at cross-polarization (CP) contact times of 70 µs (**A**), 1000 µs (**B**), and 8000 µs (**C**) acquired at a temperature of 30 °C (NMR spectra are plotted to scale). Experimental NMR spectra are shown in black and simulations of the spectral line shape in red. The areas of the deconvoluted isotropic and anisotropic backbone signals are plotted as a function of the CP contact time in (**D**). These intensities were fitted to the *I–S* model illustrated as solid lines [48].

**Figure 2 molecules-25-05489-f002:**
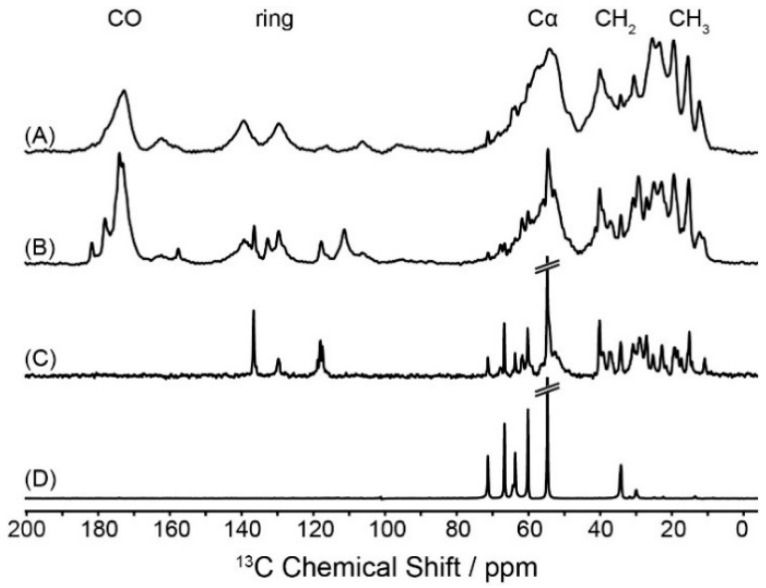
Proton-decoupled ^13^C magic-angle spinning (MAS) NMR spectra of U-^13^C-Y1R reconstituted in DMPC-*d*_54_ membranes using different polarization schemes. (**A**) ^13^C CPMAS NMR spectrum using a CP contact time of 700 µs. (**B**) Directly excited ^13^C MAS NMR spectrum. (**C**) ^13^C INEPT NMR spectrum and (**D**) ^13^C INEPT NMR spectrum of DMPC-*d*_54_ membranes in the absence of Y1R. All NMR spectra were acquired at 30 °C at a MAS frequency of 7 kHz.

**Figure 3 molecules-25-05489-f003:**
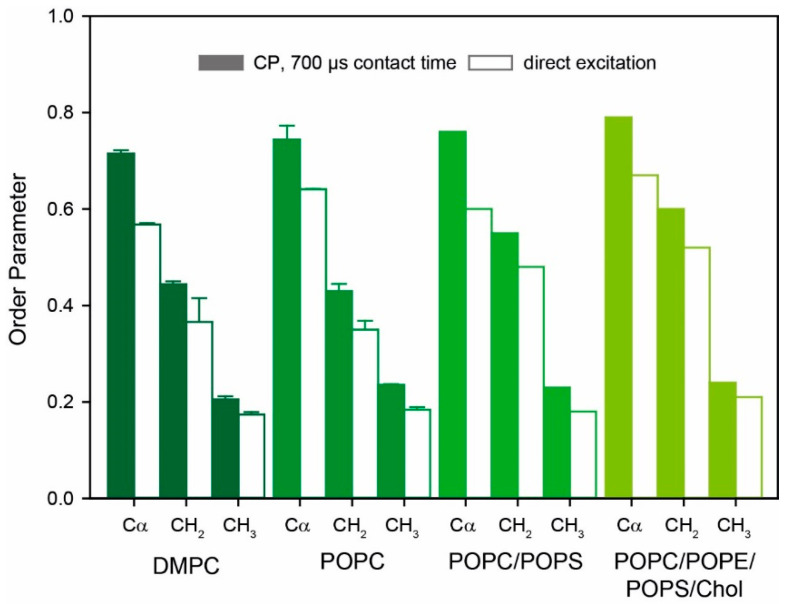
DipShift ^1^H-^13^C order parameters of the U-^13^C Y1R in DMPC, POPC, POPC/POPS (8/2, mol/mol), and POPC/POPE/POPS/cholesterol (4/4/1/1 mol/mol/mol/mol) membranes detected by either CP excitation with a 700 µs contact time (filled bars) or direct excitation (open bars). The order parameter for Cα, CH_2_, and CH_3_ groups are shown. Error bars for measurements in DMPC and POPC were determined from two independent preparations, whereas only single preparations were made for the other membrane systems.

**Figure 4 molecules-25-05489-f004:**
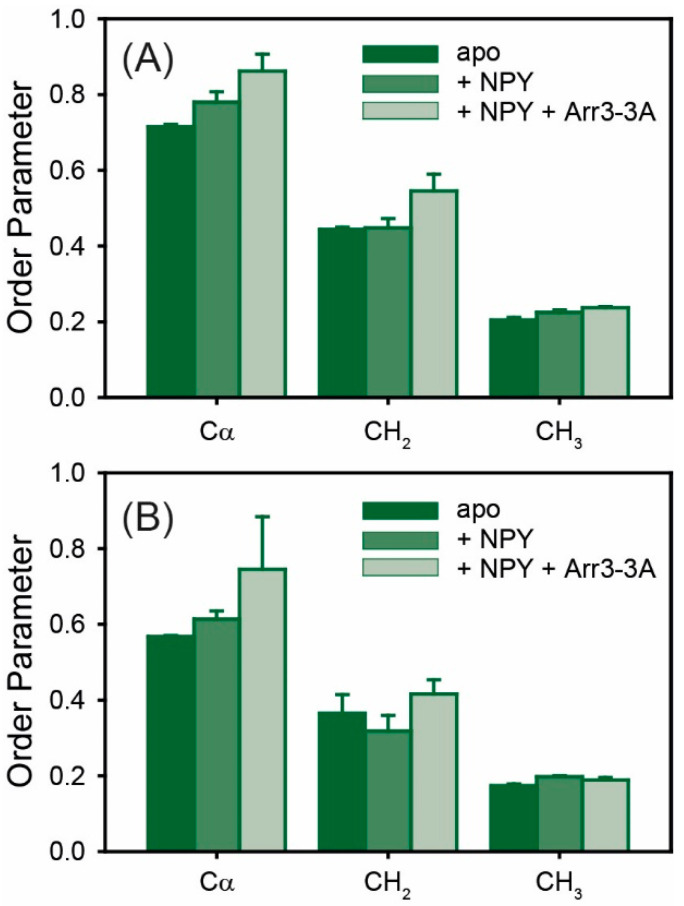
DipShift ^1^H-^13^C order parameters of Y1R in the apo, NPY-bound state, or in complex with NPY and Arr3-3A determined by DipShift experiment using CP at a contact time of 700 µs (**A**) or direct excitation (**B**). Error bars represent the experimental error determined from two independent preparations and measurements.

**Figure 5 molecules-25-05489-f005:**
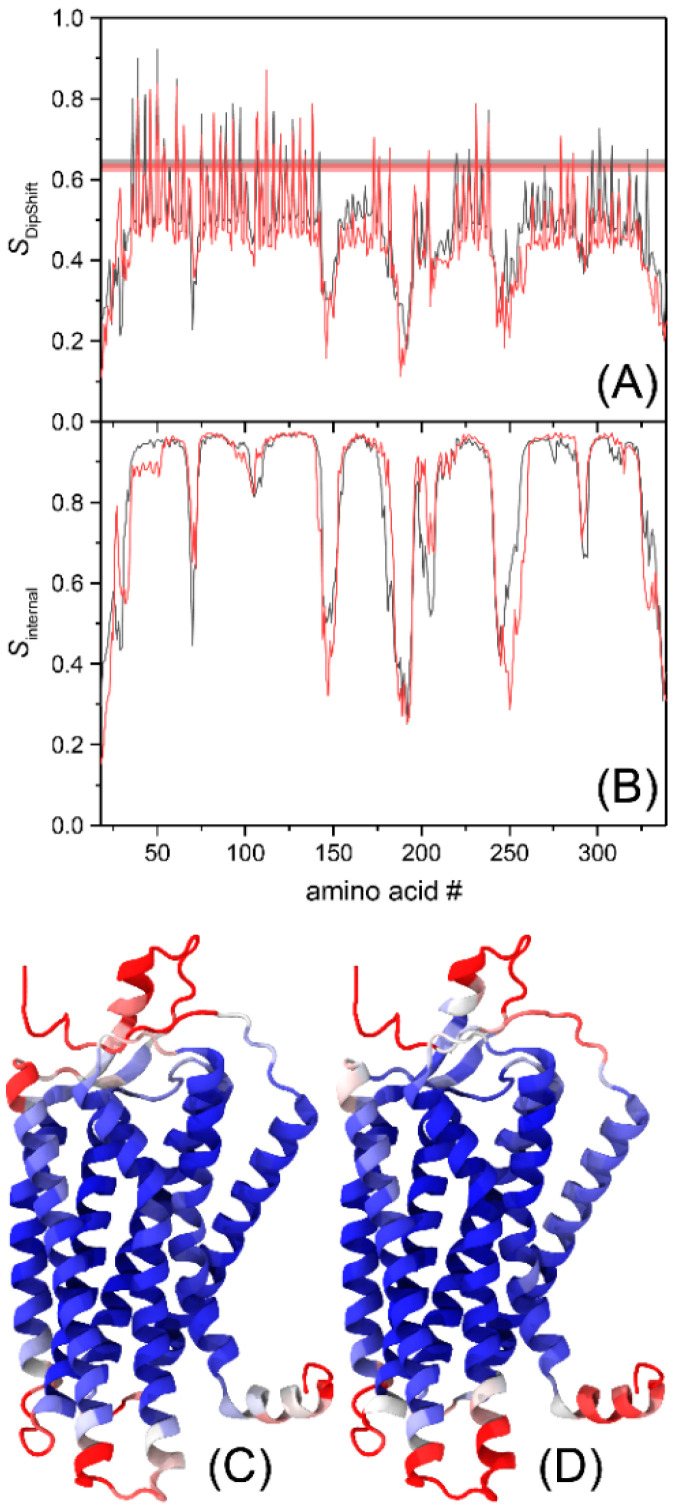
Segmental ^1^H-^13^C order parameters determined from the molecular dynamics (MD) simulations. In panel (**A**), the order parameters (arithmetic mean over all three runs) as directly observed in the MD simulations are shown (black: Y1R in the apo state, red: Y1R in the presence of UR-MK299). The experimental values (red and black lines corresponding to the values from Table 2, including the error intervals) are shown for comparison. In panel (**B**), the same analysis was performed on the trajectories, where the overall reorientation of the receptor was removed such that the order parameter now corresponds to the internal order parameter *S*_internal_. In panels (**C**) (apo state) and (**D**) (UR-MK299-bound state), the internal order parameter *S*_internal_ was projected on the Y1R structure using a color scale reaching from 0.5 (red) to 1.0 (blue).

**Figure 6 molecules-25-05489-f006:**
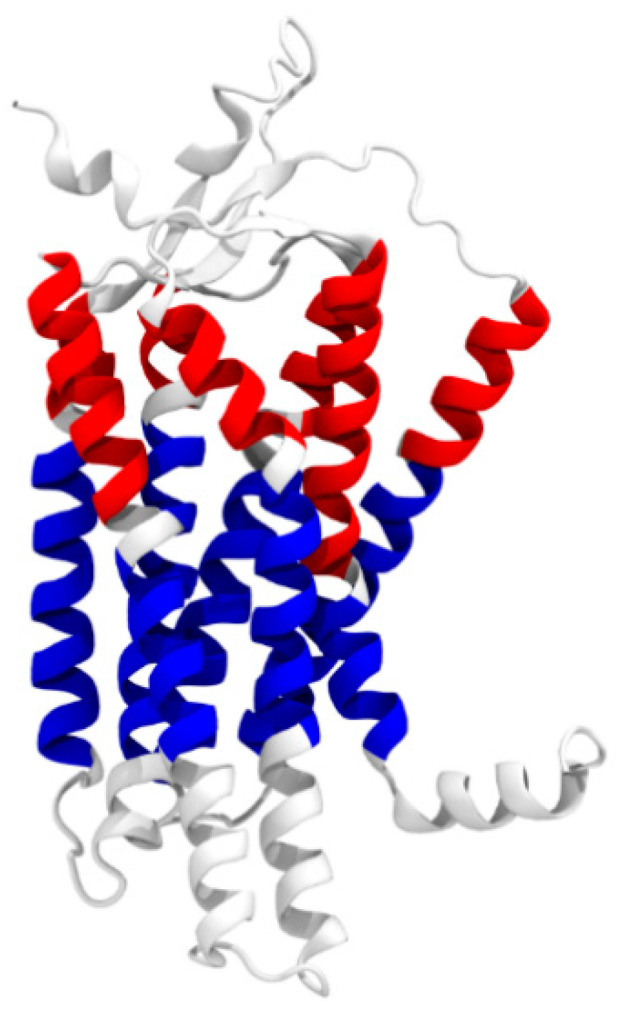
Visualization of the individual extracellular (red) and intracellular (blue) helix segments.

**Figure 7 molecules-25-05489-f007:**
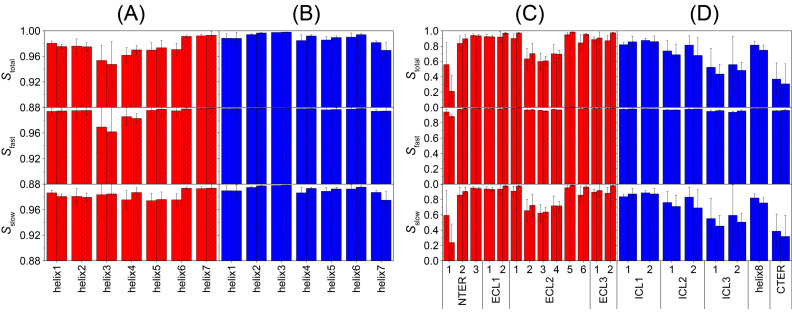
Order parameters of the individual segments of Y1R calculated from the MD simulations. The top row shows the total order parameter *S*_total_ of each segment, whereas the two lower panels show the individual contributions from fast (*S*_fast_) and slow (*S*_slow_) motions. Open bars correspond to the apo state, whereas striped bars correspond to the antagonist-bound state. Panel (**A**) corresponds to the extracellular helix segments, whereas (**B**) shows their intracellular counterparts. Panels (**C**,**D**) show the loops, termini, and Helix 8, also grouped into extracellular and intracellular segments, respectively. In all panels, the arithmetic mean over all three runs is shown with error bars corresponding to the standard deviation. Note the difference in the range of order parameters between helices (**A**,**B**) and loops, termini, and Helix 8 (**C**,**D**).

**Figure 8 molecules-25-05489-f008:**
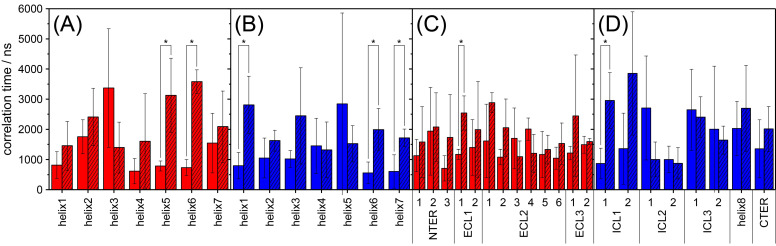
The correlation times of the individual segments that were obtained by fitting the ACFs extracted from the MD simulations. Open bars correspond to the apo state, whereas striped bars correspond to the antagonist-bound state. Panel (**A**) corresponds to the extracellular helix segments, whereas (**B**) shows their intracellular counterparts. Panels (**C**,**D**) show the loops, termini, and Helix 8, also grouped into extracellular and intracellular segments, respectively. In all panels, the arithmetic mean over all three runs is shown and the error bars correspond to the standard deviation. Statistically significant (*p* < 0.05) higher values of the antagonist-bound state are indicated by *.

**Table 1 molecules-25-05489-t001:** Molecular order parameters of Y1R reconstituted into DMPC membranes at a temperature of 30 °C in the apo state in the presence of neuropeptide Y (NPY) and in the presence of NPY and coupled to Arr3-3A.

	Cα		CH_2_		CH_3_	
Excitation Scheme	CP, 700 µs	Direct Excitation	CP, 700 µs	Direct Excitation	CP, 700 µs	Direct Excitation
Y1R (apo)	0.72 ± 0.01 *	0.57 ± 0.01	0.44 ± 0.01	0.37 ± 0.05	0.21 ± 0.01	0.18 ± 0.01
Y1R + NPY	0.78 ± 0.03	0.61 ± 0.02	0.45 ± 0.03	0.32 ± 0.04	0.23 ± 0.01	0.20 ± 0.01
Y1R + NPY + Arr3-3A	0.86 ± 0.05	0.75 ± 0.14	0.55 ± 0.04	0.42 ± 0.04	0.24 ± 0.01	0.19 ± 0.01

* Experimental errors were determined from two independent preparations.

**Table 2 molecules-25-05489-t002:** Molecular order parameters *S*_CH_ of Y1R reconstituted into POPC, POPC/POPS (8/2, mol/mol), and POPC/POPE/POPS/cholesterol (4/4/1/1, mol/mol/mol/mol) membranes at a temperature of 37 °C.

	Cα		CH_2_		CH_3_	
Excitation Scheme	CP, 700 µs	Direct Excitation	CP, 700 µs	Direct Excitation	CP, 700 µs	Direct Excitation
POPC	0.74 ± 0.03 *	0.64 ± 0.01	0.43 ± 0.02	0.35 ± 0.02	0.24 ± 0.01	0.18 ± 0.01
POPC/POPS (8/2, mol/mol)	0.76	0.60	0.55	0.48	0.23	0.18
POPC/POPE/POPS/cholesterol (8/2, mol/mol)	0.79	0.67	0.60	0.52	0.24	0.21
POPC + UR-MK299	0.76 ± 0.02	0.63 ± 0.01	0.42 ± 0.01	0.34 ± 0.01	0.23 ± 0.02	0.18 ± 0.01

* Experimental errors for measurements in POPC were determined from two independent preparations. Only single preparations were made for the other membrane systems.

**Table 3 molecules-25-05489-t003:** Summary of the amino acids that comprise the individual segments used in the analysis of the MD simulations.

Structural Element	Subs-Segment	Amino Acids
NTER	1	18–24
	2	24–30
	3	30–36
Helix 1	Extracellular	38–47
	Intracellular	49–67
ICL1	1	68–70
	2	70–73
Helix 2	Intracellular	75–91
	Extracellular	95–103
ECL1	1	104–106
	2	106–109
Helix 3	Extracellular	111–116
	Intracellular	119–138
ICL2	1	144–147
	2	147–151
Helix 4	intracellular	153–170
	Extracellular	172–176
ECL2	1	177–181
	2	181–186
	3	186–191
	4	191–196
	5	196–201
	6	201–204
Helix 5	Extracellular	205–218
	Intracellular	223–233
ICL3	1	245–248
	2	248–251
Helix 6	Intracellular	264–277
	Extracellular	279–288
ECL3	1	289–292
	2	292–295
Helix 7	Extracellular	297–314
	Intracellular	316–323
Helix 8	-	326–334
CTER	-	335–339

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
