# Peer review of "The Dynamics of the Neuropeptide Y Receptor Type 1 Investigated by Solid-State NMR and Molecular Dynamics Simulation"

_molecules, 2020, doi:10.3390/molecules25235489_

Round 1

Reviewer 1 Report

The authors investigate the dynamics of the neuropeptide Y G-protein coupled receptor (Y1R) using NMR spectroscopy as well as molecular dynamics. They measure the molecular order parameters of different areas of the protein (backbone, sidechain) and regions (loops/TM helices) using solid state NMR spectroscopy. And also characterize the order parameters in the apo., agonist, and arrestin bound states, as well as in different membrane mimetics. The authors complement this data by carrying out molecular dynamics simulation of Y1R in the absence and presence of an antagonist. Key findings are that the dynamics of the intracellular regions of Y1R are distinct from the extracellular regions. Also the addition of antagonists substantially influences the dynamics of Y1R. The manuscript is well written and technically sound, however the discussion largely focuses on explaining the results and does not make the significance of their findings to the broader field very clear.

Major comments:

-The difference between the data in Table 1 and Figure 4 is not clear. It looks like the data in the table may be simple replotted in the figure, which is very redundant.

-Page 16/Figure 8. The uncertainty of the data is quite high, so it is unclear if the authors can make the statement that the apo state has shorter correlations than the antagonist-bound state. It would be good to annotate the graphs so that one knows which data are significantly different.

-Top of page 17, it would be good to add error measurements to these ranges, so that the significance is more apparent.

- The molecular dynamics simulation run time is sometimes described as being 20 µs long, and other times as a total of over 100 µs (if all of the runs are added together). It would be more accurate to consistently state the length of the individual MD simulations, instead of the total length.

Minor comments:

Page 8: “to help identifying”

Page 8: “DipShift spectra acquired”

Page 17: near the bottom “Oder parameters”

Page 22: “LLC to allow including the palmitoylation”

Author Response

The authors investigate the dynamics of the neuropeptide Y G-protein coupled receptor (Y1R) using NMR spectroscopy as well as molecular dynamics. They measure the molecular order parameters of different areas of the protein (backbone, sidechain) and regions (loops/TM helices) using solid state NMR spectroscopy. And also characterize the order parameters in the apo., agonist, and arrestin bound states, as well as in different membrane mimetics. The authors complement this data by carrying out molecular dynamics simulation of Y1R in the absence and presence of an antagonist. Key findings are that the dynamics of the intracellular regions of Y1R are distinct from the extracellular regions. Also the addition of antagonists substantially influences the dynamics of Y1R. The manuscript is well written and technically sound, however the discussion largely focuses on explaining the results and does not make the significance of their findings to the broader field very clear.

We thank the reviewer for his/her constructive remarks. We have tried to improve the discussion to highlight the significance a bit better.

Major comments:

-The difference between the data in Table 1 and Figure 4 is not clear. It looks like the data in the table may be simple replotted in the figure, which is very redundant.

Yes, this is true, the data in the diagrams are identical to the data in the table. The rational is that diagrams are more instructive for a quick overview, but some people do appreciate having the numbers, for instance for the direct comparison to other molecules, similar measurements etc. Since Molecules is an online journal, we refrained from submitting a supplementary file as page numbers should not be a concern. If the editor suggests to move the tables into a SI file, we’d be happy to do so.

-Page 16/Figure 8. The uncertainty of the data is quite high, so it is unclear if the authors can make the statement that the apo state has shorter correlations than the antagonist-bound state. It would be good to annotate the graphs so that one knows which data are significantly different.

We evaluated the p-values for each segment in figure 8 for the hypothesis that the correlation time of the antagonist-bound state is higher than that of the apo state. Each p-value below 0.05 was considered significant. Significantly higher correlation times of the antagonist-bound state were found for 2 (out of 7) extracellular and 3 (out of 7) intracellular helix segments. For loops, termini and helix 8 statistically significant higher correlation times of the antagonist-bound state are observed for 1 (out of 13) extracellular and 1 (out of 8) intracellular segments. Fig. 8 has been updated accordingly.

-Top of page 17, it would be good to add error measurements to these ranges, so that the significance is more apparent.

Calculating the standard deviation seems not helpful in this case as the individual segments in some cases are rather different in size resulting in very different correlation times which leads to very large apparent standard deviations. Instead, we conducted pair-sample t-tests (using OriginPro 2017) where individual segments and runs were paired with each other (e.g. run1 of a helix segment in the apo state vs. run1 of the same segment in the antagonist-bound state) which eliminates this problem. For the helices, this resulted in a highly significant result (p < 0.01) for the hypothesis that the correlation time of the antagonist-bound state is higher than that of the apo state. For loops, termini and helix 8 also significantly (p < 0.05) higher correlation times of the antagonist-bound state are obtained using the same method. The manuscript has been updated accordingly.

- The molecular dynamics simulation run time is sometimes described as being 20 µs long, and other times as a total of over 100 µs (if all of the runs are added together). It would be more accurate to consistently state the length of the individual MD simulations, instead of the total length.

This has been changed as the reviewer requested.

Minor comments:

Page 8: “to help identifying”

Changed into: “to help identify”

Page 8: “DipShift spectra acquired”

Changed into: “13C DipShift spectra were acquired”

Page 17: near the bottom “Oder parameters”

Corrected.

Page 22: “LLC to allow including the palmitoylation”

Changed into: “to include the palmitoylation”

Reviewer 2 Report

The manuscript presents a very interesting study on the changes in dynamics  Y1R in different membranes. Besides their biological relevance the study is a very good example on how NMR data and MD simulations can be jointly used to bring very specific information of local and segmental molecular motions. The manuscript is well written and the results and analysis are sound. I just have a few technical remarks.

  • The authors show a quantification on the amount of mobile sites using static CP 15N NMR spectra. While the take a good care on evaluating que CP profile to avoid bias on the quantification due to the CP excitation, they do not mention about the possibility of a fraction of segments present mobility in the intermediate regime (motion correlation times of tens to hundreds microseconds) . The signals of such segments may be strongly affected either by the CP excitation or the 1H decoupling. Indeed, the 15N static spectra seems to be dynamically broadened, as revealed by the shoulder in the 2500-250 ppm region. Is this a sign of segments occurring with motion rates in the intermediate regime? In what extend the presence of such segments would affect the quantification of mobile segments presented by the authors?  Is there any special reason for the authors not consider a more real distribution of motion rates?
  • Molecular order parameters obtained from Dipshift experiments are used to probe dynamic differences in of the Y1R in different membranes. Molecular order of about 7 (as shown typically for backbone segments in figure 3) indicated that the segments are far from being rigids. It is known that both the rate and the amplitude of motions affect the DIPSHIFT modulation curves (from which molecular order parameter are extracted). Could the difference between the molecular order parameter between the NMR and MD simulations associated to the presence of a distribution of motional rates?
  • How does the authors evaluated the rigid limit dipolar coupling for estimating the molecular order parameters from Dipshift? Was it based on the expected theoretical values or in a measurement of a external reference sample? (both approaches are found in the literature). In what extend this procedure can affect the comparison between the MD simulation and NMR results?

Author Response

The manuscript presents a very interesting study on the changes in dynamics  Y1R in different membranes. Besides their biological relevance the study is a very good example on how NMR data and MD simulations can be jointly used to bring very specific information of local and segmental molecular motions. The manuscript is well written and the results and analysis are sound. I just have a few technical remarks.

We thank the reviewer for the positive evaluation of our work.

  • The authors show a quantification on the amount of mobile sites using static CP 15N NMR spectra. While the take a good care on evaluating que CP profile to avoid bias on the quantification due to the CP excitation, they do not mention about the possibility of a fraction of segments present mobility in the intermediate regime (motion correlation times of tens to hundreds microseconds) . The signals of such segments may be strongly affected either by the CP excitation or the 1H decoupling. Indeed, the 15N static spectra seems to be dynamically broadened, as revealed by the shoulder in the 2500-250 ppm region. Is this a sign of segments occurring with motion rates in the intermediate regime? In what extend the presence of such segments would affect the quantification of mobile segments presented by the authors?  Is there any special reason for the authors not consider a more real distribution of motion rates?

That is very true. Motions in the µs correlation time window may very well contribute and decrease the spectral intensity due to interference with the angular frequencies of decoupling or excitation fields. With the current dataset, we cannot account for such motions. We can only simulate/analyze spectral components that are observable in the NMR spectra. We added the following statement about possible signal losses due to µs time scale motions.

“Our analysis only considers motions with correlation times faster than ~70 µs. It is possible that slower µs timescale motions that interfere with the decoupling or excitation frequencies could lead to signal losses in the 15N NMR spectra [49]. Such motions could not be accounted for in our analysis.”

However, our 1H-13C Dipshift dephasing curves did not support the notion that µs timescale motions could play a significant role (vide infra).

  • Molecular order parameters obtained from Dipshift experiments are used to probe dynamic differences in of the Y1R in different membranes. Molecular order of about 7 (as shown typically for backbone segments in figure 3) indicated that the segments are far from being rigids. It is known that both the rate and the amplitude of motions affect the DIPSHIFT modulation curves (from which molecular order parameter are extracted). Could the difference between the molecular order parameter between the NMR and MD simulations associated to the presence of a distribution of motional rates?

We totally agree, a 1H-13C bond vector with an order parameter of 0.4 (which means S2 = 0.49) is not to be considered a rigid site. We went through the manuscript to make sure this observation is well represented in the discussion.

With regard to the correlation time, a good indication for the presence of slower motions is the observation that dipolar dephasing curves observed over one MAS rotor period do not reach the full echo maximum. This was, however, not observed. So we conclude that µs motions are not a dominating factor in the mobility of the Y1R. Some explanation is added to the discussion section.

Finally, it is very likely that the molecular motions of a GPCR in lipid membranes are subject to possibly broad distributions of the correlations times. However, given the small amount of experimental data, it is not useful to assume distribution functions in the determination of order parameters as this would add more fitting parameters to an already very sparse data set. We are currently also measuring relaxation times and hope that this additional data will allow us to better and more comprehensively describe the mobility of GPCRs in membranes. We consider the agreement between NMR and MD rather good, though, taking into account that despite our efforts, sampling is likely still incomplete. This means, reorientation of the receptor as a whole is a rather slow process and probably not sampled completely. As order parameters determined from the MD simulations are lower than the experimental values over the whole sequence, it is likely that such processes are responsible for the difference instead of the motions of individual segments which also occur on faster time scales and therefore are sampled better. With further NMR investigations (e.g. with selective labeling of individual amino acids) and their comparison to the MD simulations the picture will clarify in the future.

  • How does the authors evaluated the rigid limit dipolar coupling for estimating the molecular order parameters from Dipshift? Was it based on the expected theoretical values or in a measurement of a external reference sample? (both approaches are found in the literature). In what extend this procedure can affect the comparison between the MD simulation and NMR results?

We used experimentally determined rigid limit values for the dipolar couplings of CH, CH2, and CH3 groups as reference values. Two references were given in the previous manuscript, but we have now added one sentence of explanation to both the results and materials and methods sections.